# Early Gut Microbiota Profile in Healthy Neonates: Microbiome Analysis of the First-Pass Meconium Using Next-Generation Sequencing Technology

**DOI:** 10.3390/children10071260

**Published:** 2023-07-22

**Authors:** Yi-Sheng Chang, Chang-Wei Li, Ling Chen, Xing-An Wang, Maw-Sheng Lee, Yu-Hua Chao

**Affiliations:** 1Department of Research and Development, AllBio Life Incorporation, Taichung 402, Taiwan; ysheng@allbiolife.com (Y.-S.C.); vit@allbiolife.com (C.-W.L.); leo@allbiolife.com (L.C.); 2Department of Pediatrics, Chung Shan Medical University Hospital, Taichung 402, Taiwan; win1227.tw@yahoo.com.tw; 3Department of Obstetrics and Gynecology, Lee Women’s Hospital, Taichung 406, Taiwan; kso579@hotmail.com.tw; 4School of Medicine, Chung Shan Medical University, Taichung 402, Taiwan; 5Department of Clinical Pathology, Chung Shan Medical University Hospital, Taichung 402, Taiwan

**Keywords:** gut microbiome, meconium, neonate, next-generation sequencing, 16S rRNA sequencing

## Abstract

Gut microbiome development during early life has significant long-term effects on health later in life. The first-pass meconium is not sterile, and it is important to know the initial founder of the subsequent gut microbiome. However, there is limited data on the microbiota profile of the first-pass meconium in healthy neonates. To determine the early gut microbiota profile, we analyzed 39 samples of the first-pass meconium from healthy neonates using 16S rRNA sequencing. Our results showed a similar profile of the microbiota composition in the first-pass meconium samples. *Pseudomonas* was the most abundant genus in most samples. The evenness of the microbial communities in the first-pass meconium was extremely poor, and the average Shannon diversity index was 1.31. An analysis of the relationship between perinatal characteristics and the meconium microbiome revealed that primigravidae babies had a significantly higher Shannon diversity index (*p* = 0.041), and the *Bacteroidales* order was a biomarker for the first-pass meconium of these neonates. The Shannon diversity index was not affected by the mode of delivery, maternal intrapartum antibiotic treatment, prolonged rupture of membranes, or birth weight. Our study extends previous research with further characterization of the gut microbiome in very early life.

## 1. Introduction

The human gut is home to a wide variety of microbes termed the gut microbiome. The complex ecosystem consists of an amazing 100 trillion bacteria, and its genome represents more than 100 times the human genome [1]. A growing body of evidence indicates the importance of microbiota commensals in human health. Numerous studies have been conducted on the relationship between the gut microbiome and the occurrence of various diseases in children [2,3,4,5] and adults [6,7,8,9,10,11].

The healthy human fetus is thought to grow in a nearly sterile environment. During the perinatal period, the gut is initially colonized by diverse microbes, mainly provided by the mother and the surrounding environment [12,13]. Thereafter, the gut microbiome undergoes a maturation process with great shifts in both composition and diversity. A mature gut microbiome is securely established in an adult form by 3 years of age. Several studies on the developmental trajectories demonstrated that the gut microbiome undergoes most of its development very early in life, with the strongest changes occurring during the first year of life [14,15,16,17]. Meanwhile, the developmental process during early life has significant and lasting effects on health later in life [18,19,20,21]. Therefore, research on the evolution of the gut microbiota community structure during early childhood has recently received a lot of attention.

The first-pass meconium is not sterile, although there are a low number of bacteria within it [22,23,24,25,26]. The first colonizers play an important role in establishing the gut ecosystem. They also impact the microbiota composition and activity throughout life and have a significant influence on later-in-life host physiology and health [27,28,29,30]. Accordingly, identifying the reference founder from which the subsequent gut microbiome is derived is the mainstay of research on the relationship between the gut microbiome and human diseases. However, limited data are available regarding the microbial signatures of the first meconium. Therefore, we aimed to determine the early gut microbiome of the first-pass meconium from healthy neonates in the present study by using 16S rRNA sequencing technology.

## 2. Materials and Methods

### 2.1. Study Population and Sampling

From December 2022 to February 2023, all pregnant women admitted to our hospital were invited to participate in this study at the time of admission to labor and delivery. This prospective observational cohort study was approved by the Institutional Review Board of Chung Shan Medical University Hospital (CS2-22178), and written informed consent was obtained from all participants.

The first-pass meconium, i.e., the first stool after birth, was collected by the child’s named nurse in the baby room. Fresh meconium samples were obtained from diapers after spontaneous evacuation and stored in a collecting tube prefilled with 10 mL of buffer solution (AllBio Science, Taichung, Taiwan) at room temperature before being transported to the laboratory at the AllBio Life Incorporation. For negative control, control samples were collected from blank diapers to assess contamination. Clinical information related to birth was also collected.

Newborn infants who fulfilled all the following criteria were eligible for data analysis: (1) babies were born to healthy mothers after normal pregnancy; (2) the delivery process was uncomplicated and the Apgar scores were higher than 7; (3) babies were apparently normal without congenital anomalies; (4) babies did not require medical treatment; (5) babies started oral feeds within the first 24 h; and (6) the first-pass meconium was spontaneously evacuated after birth. Participants were excluded based on the following criteria: (1) inability to give informed consent; (2) significant background maternal health concerns; (3) significant maternal health issues during pregnancy, such as gestational diabetes mellitus and hypertension; (4) neonatal health concerns sufficient to warrant admission to the neonatal unit; and (5) delayed passage of meconium (i.e., >48 h after birth).

### 2.2. DNA Extraction and Sequencing

Total genome DNA from meconium samples was extracted using the EasyPure Stool Genomic DNA Kit (AllBio Science, Taichung, Taiwan), and DNA concentration was determined using the Qubit dsDNA HS Assay Kit (Thermo Fisher Scientific, Waltham, MA, USA). The samples were stored at −20 °C for preservation before further PCR amplification and sequencing.

A sequencing library was constructed using the MetaVx Library Preparation Kit (Genewiz, South Plainfield, NJ, USA). The V3 and V4 hypervariable regions of prokaryotic 16S rDNA were selected to generate amplicons for taxonomic analysis. The forward primer sequence was ‘CCTACGGRRBGCASCAGKVRVGAAT’ and the reverse primer sequence was ‘GGACTACNVGGGTWTCTAATCC’. In the second-stage PCR, adapters and index sequences were added to either end of the amplified fragment. The library was purified using magnetic beads, and the concentration was determined using a microplate reader (Tecan Infinite 200 Pro). The fragment size was determined by agarose gel electrophoresis. Next-generation sequencing was conducted using the Illumina MiSeq Platform (Illumina, San Diego, CA, USA). Automated cluster generation and 250/300 paired-end sequencing with dual reads were performed, according to the manufacturer’s instructions.

### 2.3. Bioinformatic Analysis

The sequencing data were analyzed using QIIME2 (version 2021.11) [31]. The DADA2 algorithm was used for quality filtering and merging sequences with greater than 97% similarity. The taxonomic assignment of each operational taxonomic unit (OTU) was performed according to the Greengenes database, generating a table of amplicon sequence variants for further analysis. After taxonomic classification, random sampling was applied to flatten the number of sequences in all the samples. The taxa of the same type were agglomerated at the phylum, class, order, family, genus, and species levels.

Biodiversity was calculated at the OTU level using the R software (version 4.3.0) and plotted using the MicrobiotaProcess package. Alpha diversity was used to investigate species richness and evenness within samples, as illustrated by the Shannon index. For beta diversity, phylogeny-based UniFrac analysis was performed to evaluate the diversity and degree of difference among samples using the PERMANOVA method. Principal components analysis (PCA) was based on the taxonomic distribution; principal coordinates analysis (PCoA) and non-metric multidimensional scaling (NMDS) were based on the Bray–Curtis distance matrix. A linear discriminant analysis of effect size (LEfSe) was applied to determine the taxa that best discriminated between the groups of samples. LEfSe was used for differential abundance analysis to generate linear discriminant analysis (LDA) plots.

## 3. Results

### 3.1. Study Cohort

A total of 39 neonates fulfilled all the inclusion criteria and were enrolled for data analysis. There were 19 males and 20 females. The clinical data are outlined in Table 1.

### 3.2. Distinct Microbiota Community Structures in the Meconium Samples and Control Samples

We sequenced and analyzed 39 samples of the first-pass meconium from healthy neonates and 3 samples from blank diapers. Sequencing generated an average of 187,873 reads per sample, and 3067 OTUs were identified. Because it is impossible to obtain meconium via a sterile procedure, samples collected from blank diapers were used for sequencing control to assess contamination.

Beta diversity was calculated to evaluate the differences in the microbiota community structure between the first-pass meconium and the control. As shown in Figure 1A, PCA of the predicted microbial functions revealed that the meconium samples were concentrated and distinct from the control samples. The Bray–Curtis distance-based similarity analysis by PCoA also indicated that the meconium samples were clustered separately (Figure 1B). Concordantly, NMDS analysis demonstrated a significant difference in the distance between the two groups (Figure 1C).

The Venn diagram display the number of OTUs that were shared or exclusive, and the first-pass meconium and the control only shared 9.3% of the OTUs (Figure 2A). Consistently, the heatmap at the phylum level revealed that the meconium samples and control samples had distinct microbiota community structures (Figure 2B).

### 3.3. A Similar Profile of the Microbiota Composition in the First-Pass Meconium

Next, a microbiome analysis of the first-pass meconium was performed. The taxonomic analysis identified the top 30 OTUs with the highest abundance at the genus level, and a similar profile of the microbiota composition in the first-pass meconium samples was observed (Figure 3). Notably, *Pseudomonas* was the most abundant genus in most samples, accounting for >80% of all sequences. The relative abundances of *Pseudomonas* were less than 50% in only five meconium samples. All five neonates evacuated the first-pass meconium significantly later, 24 h after birth.

As the common composition profile of the first-pass meconium was noted, the predominant bacteria at each taxonomic level were further identified. The average abundances of the top five OTUs in the meconium samples from phylum to species level are presented in Table 2. *Proteobacteria* and *Firmicutes* were the two most abundant phyla in the first-pass meconium, accounting for an average abundance of 85.45% and 9.74%, respectively. At the species level, *Pseudomonas* sp. (EU557337) was dominant with an average abundance of 17.35%. Of importance, the evenness of the microbial communities in the first-pass meconium was very poor. From phylum to genus level, most of the bacterial population belonged to a single taxon. The average Shannon diversity index was 1.31.

### 3.4. Relationship between Perinatal Characteristics and the First-Pass Meconium Microbiome

The relationship between perinatal characteristics and the first-pass meconium microbiome was evaluated. The effects of gravidity, mode of delivery, maternal intrapartum antibiotic treatment, prolonged rupture of membranes, and birth weight on microbiota diversity of the first-pass meconium were determined. As shown in Figure 4, primigravidae babies had a significantly higher Shannon diversity index than babies of multiparous mothers (*p* = 0.041). Using LEfSe to identify the most differentially abundant taxa, the *Bacteroidales* order (the *Bacteroidia* class) was found to be a biomarker for the first-pass meconium of these neonates (LDA score ≥ 4). Meanwhile, the Shannon diversity index was not affected by the mode of delivery, maternal intrapartum antibiotic treatment, prolonged rupture of membranes, or birth weight.

## 4. Discussion

Information on the microbiota community structure of the first-pass meconium in healthy neonates is important to identify the reference founder from which the subsequent gut microbiome is derived. However, limited data are available in the literature. In the present study, we analyzed 39 samples of first-pass meconium from healthy neonates using 16S rRNA sequencing. Our results showed a similar profile of the microbiota composition in the first-pass meconium samples in healthy neonates. The predominant bacteria at each taxonomic level were identified, and *Pseudomonas* was found to be the most abundant genus in most samples. The evenness of the microbial communities was extremely poor in the first-pass meconium. Our study extends previous research with further characterization of the gut microbiome in very early life.

The human gut acquires a nascent microbiome at, or perhaps even before, the time of birth. The initial microbes mainly come from the mother and the surrounding environment during the perinatal period [12,13]. Thereafter, the gut microbiome undergoes a maturation process. The gut microbiome undergoes most of its development very early in life [14,15,16,17]. In the present study, most of the neonates had a similar profile of the microbiota composition in their first-pass meconium, implicating the pivotal role of the environment in establishing the initial meconium microbiome. Meanwhile, we observed that the meconium microbiota profiles of five neonates were apparently different from others. Of interest, all these neonates started oral feeds early and evacuated the first-pass meconium after 24 h of age. This suggested that a great shift in the gut microbiome occurred soon after birth and that infant food provided an important source of microbes for organizing the gut microbiome in early life.

The gut microbiome evolves within a healthy host from birth to death [32]. In addition to shifts in the microbiota composition, the maturation process of the infant’s gut microbiome tends to be a more complex and dense microbial colonization. As an indicator of the microbial richness and evenness of the gut ecosystem, alpha diversity was strongly associated with the development of a mature microbiome and was found to sequentially increase as children grew older [15,16,17]. Although the gut microbiota can acquire an adult-like configuration at 5 years old, there is still a lower community richness with some missing critical taxa of the adult microbiota [15]. In the present study, the evenness of the microbiota community structure in the first-pass meconium was strikingly poor. Most of the bacterial population in the first-pass meconium samples belonged to a single taxon, from phylum to genus. The average Shannon diversity index was only 1.31. Our data suggested that the human gut microbiome initiated the developmental process from a few small groups of certain microbes.

While the gut microbiota structure in healthy children has been somewhat characterized, limited data are available on the predominant bacteria at each taxonomic level. In this study, we described the average abundances of the top five OTUs in the first-pass meconium samples from phylum to species level. In 2022, El Mouzan et al. characterized gut microbiota profiles in 20 healthy school children and reported the average relative abundances of the dominant organisms from phylum to species level [33]. As expected, their profiles were quite different from ours because of a significant difference in the ages of the participants. Furthermore, an individual’s gut microbiome is shaped by multiple factors, including the host and the environment [32,34,35,36,37,38]. Consequently, there are variations in both composition and diversity from population to population [39]. A variety of perinatal and postnatal factors involving the mother–child symbiosis have been proposed to have a significant influence on the temporal development of the gut microbiome during very early life [32]. Our data revealed a similar profile of the microbiota composition in the first-pass meconium from healthy neonates, suggesting a common developmental origin of the gut microbiome in these individuals despite different trajectories may go on.

The development of the gut microbiome during early life has significant and lasting effects on later-in-life host physiology and even on the occurrence of various diseases [18,19,20,21]. The first-pass meconium reflects the microbial environment in the perinatal period, and knowledge of the meconium microbiome in healthy neonates is the cornerstone for clinical research on the role of gut microbiome in childhood diseases. Several studies were conducted to determine the microbiota composition of the meconium using diverse techniques, including bacterial culture, PCR, fluorescent in situ hybridization, electron microscopy, and next-generation sequencing [22,23,24,25,26,40]. With the advancement of detection techniques, the important issue that meconium from healthy neonates is not actually sterile has been demonstrated [22,23,24,25,26]. However, there is no consensus regarding the origin of these microbes. By analyzing the microbiota composition, the meconium microbiota may be derived from swallowed amniotic fluid [24], the placenta, the maternal digestive tract [25], and the surrounding environment [22]. Using 16S rRNA sequencing, our data showed a similar profile of the microbiota composition in the first-pass meconium of healthy neonates, suggesting a pivotal role of the surrounding environment in establishing the initial meconium microbiome.

Our study had several limitations. First, the surrounding environment is an important source of the initial microbes. Although inter-individual variations in the environment can be minimized by enrolling participants from a single center, we cannot evaluate the environmental influence on the meconium microbiome. Second, first-pass meconium contains relatively less abundant microbes. Although adequate samples for sequencing control were used to assess contamination throughout the study, the procedure to obtain the meconium was not sterile. Finally, the early development of the gut microbiome has a significant influence on health later in life. Only the data of the first-pass meconium were provided in the present study.

## 5. Conclusions

In summary, our results showed a similar profile of the microbiota composition in the first-pass meconium of healthy neonates, with predominant, but perhaps transient, colonization of non-pathogenic commensal microbiota from the *Pseudomonas* genus. The evenness of the microbial communities in the meconium was extremely poor. Further studies are needed to confirm this observation and to elucidate the relationship between the environment and the meconium microbiome.

## Figures and Tables

**Figure 1 children-10-01260-f001:**
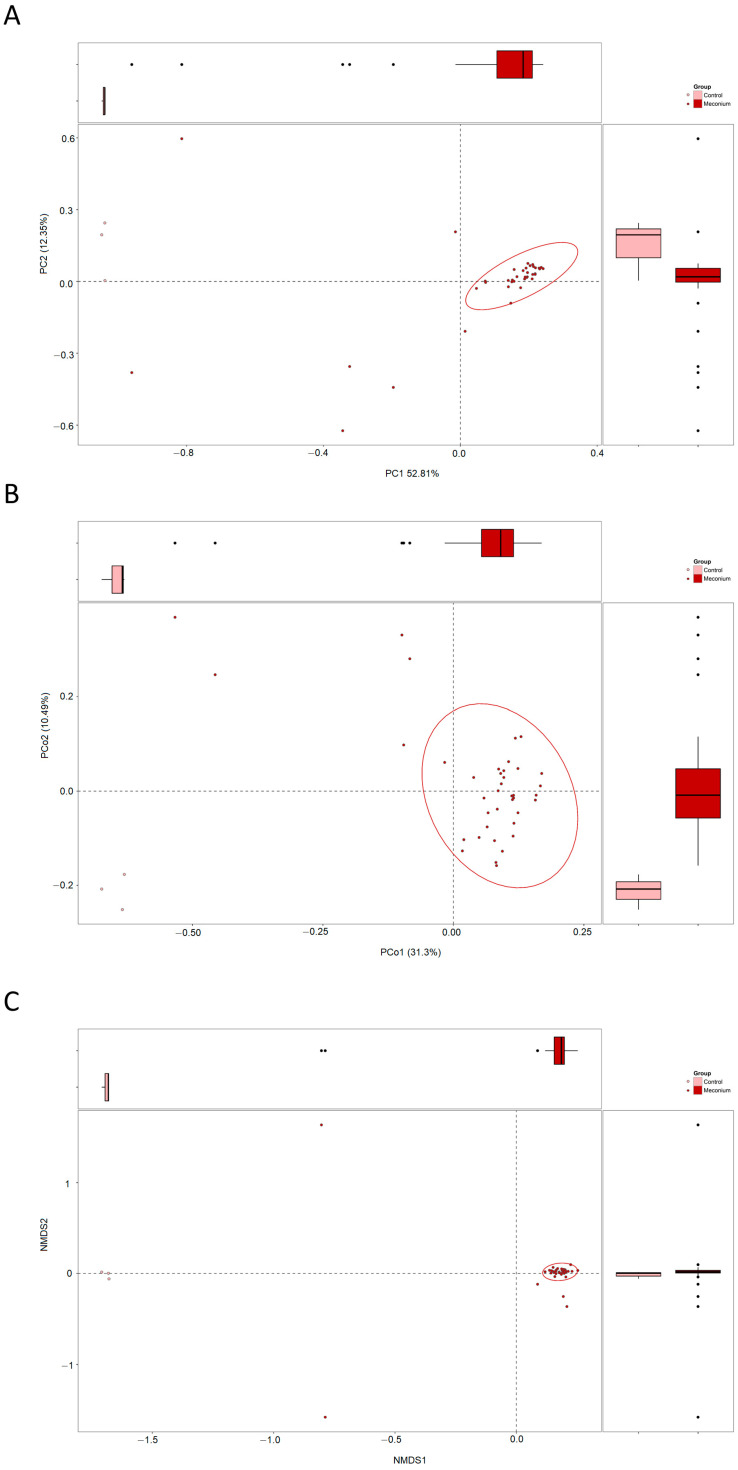
Beta diversity illustrating distinct microbiota community structures in the meconium samples and control samples. (**A**) Principal components analysis (PCA). (**B**) Principal coordinates analysis (PCoA). (**C**) Non-metric multidimensional scaling (NMDS) analysis.

**Figure 2 children-10-01260-f002:**
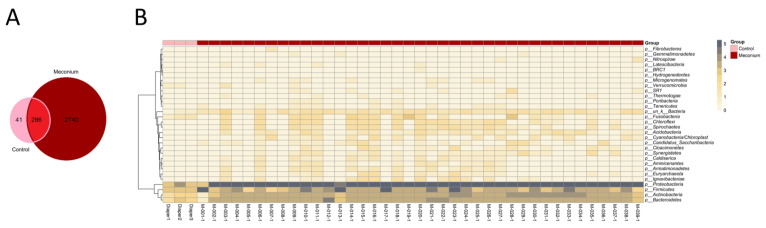
Distinct microbiota profiles of the meconium samples and control samples. (**A**) Venn diagram. (**B**) OTU heatmap at the phylum level.

**Figure 3 children-10-01260-f003:**
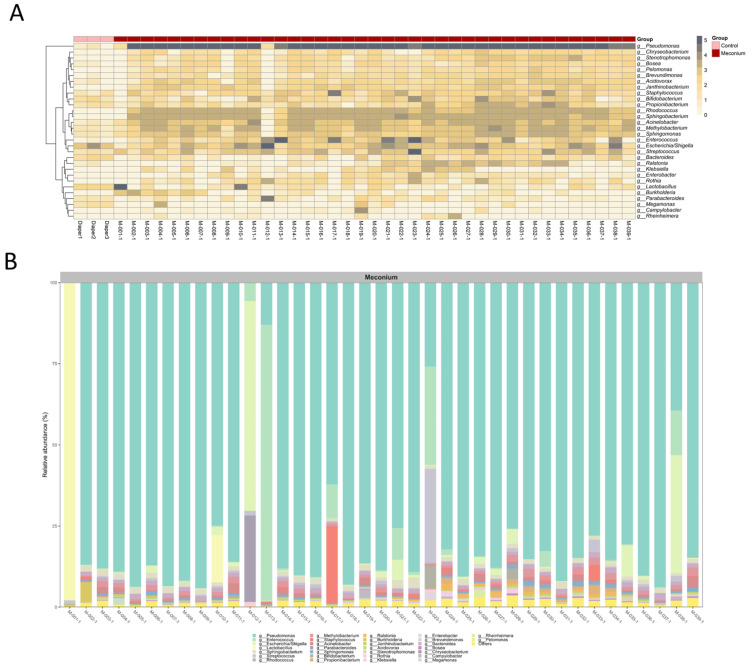
A similar profile of the microbiota composition in the first-pass meconium. (**A**) Heatmap of the top 30 OTUs at the genus level. (**B**) Bar plot showing the relative abundances of the annotated bacterial genera in each sample. The 30 most abundant genera across all samples are shown, with the remaining minor genera combined in the “Others” group.

**Figure 4 children-10-01260-f004:**
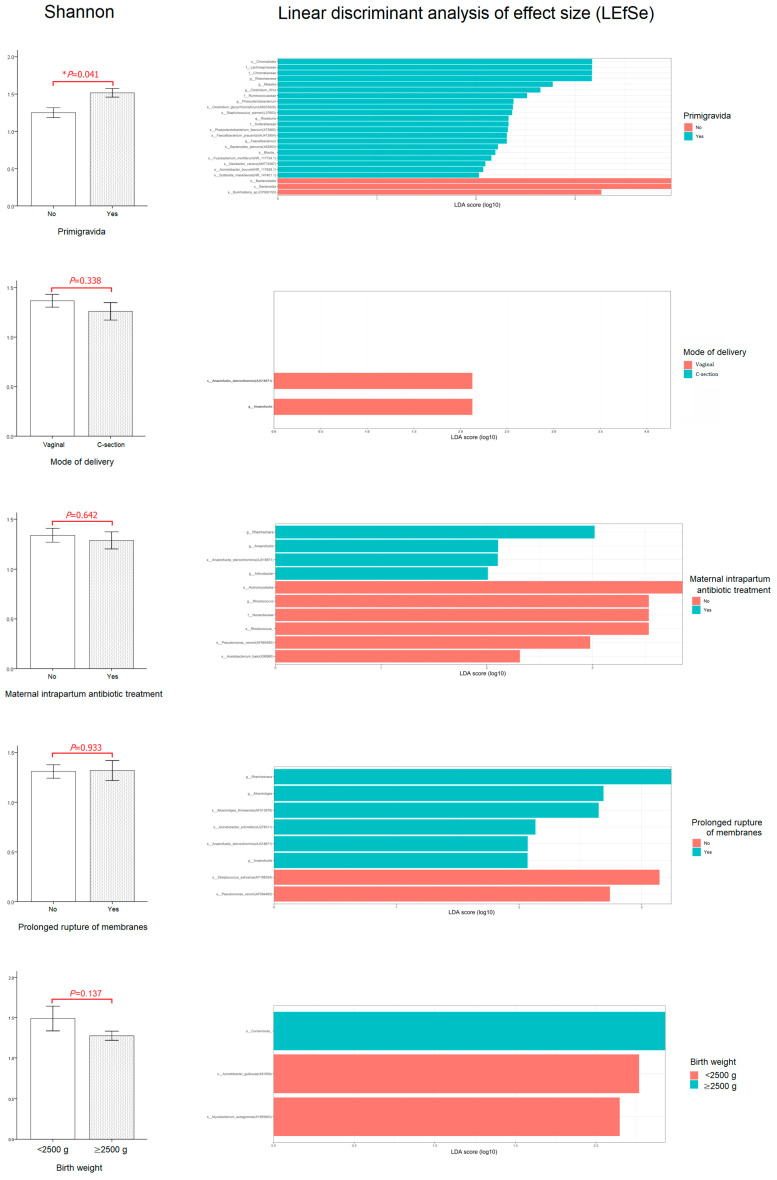
Relationship between perinatal characteristics and the first-pass meconium microbiome. Alpha diversity data are presented as mean ± SE, and a *t*-test was used for statistical analysis. * indicates *p* < 0.05.

**Table 1 children-10-01260-t001:** Descriptive data of the study population.

Characteristics	N = 39
Gender	
Male	19
Female	20
Gestational age (weeks)	38.1 (35.2–40.0)
Birth weight (g)	2880 (2020–3620)
Mode of delivery	
Vaginal	19
C-section	20
Primigravida	9 (23.0%)
Mother’s age (years)	33.8 (23.9–45.9)
Maternal intrapartum antibiotic treatment	21 (53.8%)
Prolonged rupture of membranes	12 (30.7%)
Feeding style	
Exclusive breastfeeding or formula feeding	0
Mixed (breast + formula feeding)	39 (100%)

Data are shown as numbers (percentage) and median (range).

**Table 2 children-10-01260-t002:** Meconium microbiota profile from phylum to species in healthy neonates.

Level	Organism	Abundance (%)	Level	Organism	Abundance (%)
Phylum	*Proteobacteria*	85.45	Family	*Pseudomonadaceae*	77.11
	*Firmicutes*	9.74		*Enterococcaceae*	4.30
	*Bacteroidetes*	2.42		*Enterobacteriaceae*	3.88
	*Actinobacteria*	2.25		*Lactobacillaceae*	2.91
	*Fusobacteria*	0.03		*Sphingobacteriaceae*	1.43
Class	*Gammaproteobacteria*	82.21	Genus	*Pseudomonas*	77.09
	*Bacilli*	9.44		*Enterococcus*	4.30
	*Actinobacteria*	2.25		*Escherichia/Shigella*	3.60
	*Alphaproteobacteria*	1.96		*Lactobacillus*	2.91
	*Sphingobacteriia*	1.43		*Sphingobacterium*	1.42
Order	*Pseudomonadales*	78.06	Species	*Pseudomonas sp.* (EU557337)	17.35
	*Lactobacillales*	8.49		*Streptococcus salivarius*	0.83
	*Enterobacteriales*	3.88		*Parabacteroides distasonis*	0.69
	*Actinomycetales*	1.80		*Enterococcus faecalis*	0.65
	*Sphingobacteriales*	1.43		*Methylobacterium komagatae*	0.50

## Data Availability

Sequencing datasets were deposited in the 4TU.ResearchData; URL (accessed on 3 July 2023) https://doi.org/10.6084/m9.figshare.23615784. Further inquiries can be directed to the corresponding author upon reasonable request.

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
