# Peer review of "Early Gut Microbiota Profile in Healthy Neonates: Microbiome Analysis of the First-Pass Meconium Using Next-Generation Sequencing Technology"

_children, 2023, doi:10.3390/children10071260_

Round 1

Reviewer 1 Report

In this study, the authors aimed to determine the early gut microbiome of the first-pass meconium from healthy neonates by using 16S rRNA sequencing technology.
Please specify the type of study in methods. Please also add the exclusion criteria from the study.
Tables and figures are clear and correct.
Please discuss the limitations of the study.
I recommend that you add a separate paragraph with the conclusions, they should be clear and concise.
The article presents 32 references, being up to date. You can improve their number by expanding the discussions.

Reviewer 2 Report

In the present original article Chang et al investigated the composition of gut microbiota of first-pass meconium in newborns, showing predominance of Pseudomonas and high variability.

Overall, this is a very interesting article and I would like to raise only few minor observations:

1) Page 7 lines 181-184: was birth weight included in this analysis?

2) Did some mothers experience gestational diabetes?

3) In the Discussion, comparison with other similar studies (ref. 19-22, line 241) should be more thorough.
